

# Comparison of greenhouse gas fluxes and microbial communities from tropical forest and adjacent oil palm plantations on mineral soil

Julia Drewer[1], Melissa M. Leduning[2], Robert I. Griffiths[3], Tim Goodall[3], Peter E. Levy[1], Nicholas Cowan[1], Edward Comynn-Platt[3,4], Garry Hayman[3], Justin Sentian[2], Noreen Majalap[5], Ute M. Skiba[1]

[1]UK Centre for Ecology & Hydrology, Bush Estate, Penicuik, EH26 0QB, UK

[2]Faculty of Science and Natural Resources, Universiti Malaysia Sabah, Jalan UMS, 84400 Kota Kinabalu, Malaysia

[3]UK Centre for Ecology & Hydrology, Maclean Building, Benson Lane, Wallingford, Oxfordshire, OX10 8BB, UK

[4]European Centre for Medium Range Weather Forecasting, Shinfield Road, Reading, Berkshire, RG2 9AX, UK

[5]Forest Research Centre, Sabah Forestry Department, Jalan Sepilok, Sepilok, 90175 Sandakan, Sabah, Malaysia

*Correspondence to*: Julia Drewer (juew@ceh.ac.uk)

**Abstract.** In Southeast Asia, oil palm plantations have largely replaced tropical forests. The impact of this shift in land-use on greenhouse gas (GHG) fluxes and soil microbial communities remains highly uncertain, mainly due to a relatively small pool of available data. The aim of this study is to quantify differences of nitrous oxide ($N_2O$) and methane ($CH_4$) fluxes as well as soil carbon dioxide ($CO_2$) respiration rates from logged forests, oil palm plantations of different ages and an adjacent small riparian area. The focus of this study is on $N_2O$ fluxes, as these emissions are expected to increase significantly due to the introduction of nitrogen (N) fertiliser application. This study was conducted in the SAFE





(Stability of Altered Forest Ecosystems) landscape in Malaysian Borneo (Sabah) with measurements
every two months over a two-year period. GHG fluxes were measured by static chambers; at the same
time soil samples were collected for analysis of the key soil physicochemical parameters and for analysis
of microbial biodiversity using next generation sequencing in dry and wet season. $N_2O$ fluxes were highly
variable across the different sites, with the highest mean flux from OP (46.2±166 µg m$^{-2}$ h$^{-1}$ $N_2$O-N) and
riparian (31.8±220 µg m$^{-2}$ h$^{-1}$ $N_2$O-N) sites, compared to lower fluxes from logged forest (13.9±171 µg
m$^{-2}$ h$^{-1}$ $N_2$O-N). Methane fluxes were generally small; -2.6±17.2 µg $CH_4$-C m$^{-2}$ h$^{-1}$ for OP and 1.3±12.6
µg $CH_4$-C m$^{-2}$ h$^{-1}$ for riparian with the range of measured $CH_4$ fluxes largest in logged forests (2.2±48.3
µg $CH_4$-C m$^{-2}$ h$^{-1}$). Soil respiration rates were larger from riparian areas (157.7±106 mg m$^{-2}$ h$^{-1}$ $CO_2$-C)
and logged forests (137.4±95 mg m$^{-2}$ h$^{-1}$ $CO_2$-C) than OP plantations (93.3±70 mg m$^{-2}$ h$^{-1}$ $CO_2$-C) due to
larger amounts of decomposing leaf litter. Microbial communities were distinctly different between the
different land-use types and sites, bacterial communities linked to soil pH and fungal and eukaryotic
communities to land-use. Despite measuring a number of environmental parameters, mixed models could
only explain up to 17% of the variance of measured fluxes for $N_2O$, 3% of $CH_4$ and 25% of soil respiration.
Scaling up measured $N_2O$ fluxes to Sabah using land areas for forest and OP resulted in emissions
increasing from 7.6 Mt (95% confidence interval, -3.0-22.3 Mt) per year in 1973 to 11.4 Mt (0.2-28.6 Mt)
per year in 2015 due to the increasing area of forest converted to OP plantations over the last ~40 years.





## 1 Introduction

Deforestation in Southeast Asia is so intense that up to three quarters of its forests might be lost by the end of the 21st century (Sodhi et al., 2004) and most of the degradation happens because of conversion of forest to croplands and plantations (Wilcove et al., 2013). In Malaysia and Indonesia, more than 16 million hectares of land, mainly from tropical forests but also to a lesser extent, other non-profitable agricultural land such as rubber plantations and peat, were cleared for oil palm (OP) (Yan, 2017). Many of the remaining forests are degraded forests, as they have been partially logged, to remove specific tree species and logging activity has caused an increase in forest openings (Houghton, 2012). In 20% of the world's tropical forests, selective logging occurs, and it is estimated that this accounts for at least half of the anthropogenic greenhouse gas emissions (GHG) from forest degradation (Pearson et al., 2017). Consequently, forest degradation has been recognised as a source of GHG emissions, but little is known of the emissions from the resulting secondary forests, especially in Malaysian Borneo, Sabah. Due to deforestation, fragments of forest remain isolated from each other, which can have consequences for biodiversity and ecosystem function (Ewers et al., 2011).

OP plantations are one of the main causes of deforestation and forest degradation in Southeast Asia (Lee-Cruz et al., 2013; Wilcove et al., 2013) with some disputes about the extent to which industrial plantations are responsible for the loss of old-growth and selectively logged forests in Borneo (Gaveau et al., 2016). OP generates the highest yield per hectare of land of any vegetable oil crops. It is used in food products, detergents, soaps, cosmetics, animal feed and bioenergy, and was hence praised as a wonder crop (Sayer

et al., 2012). However, OP agriculture is now known to be responsible for soil degradation, loss of soil
carbon (C) and reduced soil fertility due to the conversion and management methods (Guillaume et al.,
2015; Lee-Cruz et al., 2013). To create an OP plantation, complete deforestation followed by terracing of
the land is often the chosen method, and not only in hilly terrain. Terracing can result in poor drainage,
reduced soil fertility and increased soil erosion. Conversion of tropical forests may also lead to changes
in the short- and long-term nutrient status of the converted land-use systems. It is important to understand
impacts of these land-use changes in order to identify more environmentally friendly and sustainable
management practices (Jackson et al., 2019).

OP plantations are assessed for their GHG emissions, but rarely have emissions from forests and
plantations from the same region been reported together, despite the call to study fluxes in forest and
converted land simultaneously (van Lent et al., 2015). Much of the focus has been on GHG emissions
from peatland rather than mineral soil, either tropical forest on peatland or peatland drained for
plantations. More attention has been given to carbon fluxes or storage (Germer and Sauerborn, 2008;
Hassler et al., 2015) than emissions from the non-$CO_2$ GHG methane ($CH_4$) and nitrous oxide ($N_2O$).
Meijide *et al.* (2020) identified the need to study all three GHGs to assess total emissions from OP
plantations. Even though $CH_4$ and $N_2O$ are not emitted at the quantity of $CO_2$, their global warming
potentials (GWP) per molecule are 28 – 34 (without and with climate-carbon feedback) and 265 – 298
times higher than $CO_2$ on a 100 year time horizon, respectively, which highlights their importance (Myhre
et al., 2013).  Due to a number of environmental issues arising from conversion of peatlands to OP





plantations, the focus will increasingly shift to mineral soil for conversion to plantations, especially in
Malaysia (Shanmugam et al., 2018). There are too few measurements reported of $N_2O$ emissions from
mineral soils in the tropics to draw firm conclusions about the increase of $N_2O$ emissions after land-use
change from secondary forest to OP (Shanmugam et al., 2018).

Limited measurement and modelling studies have been carried out on $N_2O$ emissions from OP plantations
(Pardon et al., 2016a; Pardon et al., 2016b; Pardon et al., 2017), and not in the context of comparing them
with other land-uses on the same or similar soil type. Similarly, reported $CH_4$ emissions from mineral
soils in the Tropics (other than from paddy soils) are lacking. Most studies relating land-use change to
trace gas emissions have been conducted in South America and not South East Asia (Hassler et al., 2015;
Veldkamp et al., 2013). An additional caveat of published studies is that most have only been conducted
over short periods of time (Hassler et al., 2015). The lack of reliable long-term and multi-year datasets on
GHG balances has been recognised (Corre et al., 2014; Courtois et al., 2019).  Studies are often associated
with high uncertainties (Henders et al., 2015). Nitrogen availability, soil moisture and texture are the main
drivers of $N_2O$ fluxes in tropical forests and other soil ecosystems (Davidson et al., 2000). As well as
agricultural soils, tropical forest soils have been identified as a major source of $N_2O$ (Werner et al., 2007),
and soil type influences $N_2O$ fluxes in the Tropics (Dutaur and Verchot, 2007; Sakata et al., 2015). A
recent meta-analysis concluded that tropical forests emit on average 2 kg $N_2O$-N ha$^{-1}$ y$^{-1}$, and emission
rates will significantly increase after land-use change (van Lent et al., 2015). Tropical forest soils are
estimated to contribute 28% to the global $CH_4$ uptake, hence large changes to this sink could alter the



accumulation of $CH_4$ in the atmosphere substantially (Dutaur and Verchot, 2007). However, uncertainties
are large due to data scarcity. Only one study from Peninsula Malaysia reported that selectively logged
forest may be converted into a weaker sink of $CH_4$ and greater source of $N_2O$ than undisturbed tropical
rain forest, at least for a short period, because of the increased soil nitrogen availability and soil
compaction due to disturbance by heavy machinery (Yashiro et al., 2008).

Forest conversion to OP has shown differences in soil microbial community composition and functional
gene diversity (Tripathi et al., 2016). The diversity and abundance of plant communities fundamentally
affect soil microbial community and their function (Eisenhauer, 2016; Tripathi et al., 2016). As yet, it
remains uncertain how conversion from forest to OP impacts microbial communities, and their influence
on $N_2O$ and $CH_4$ fluxes (Kaupper et al., 2019). Even though the importance of bacterial communities is
recognised, little is known of changes in microbial communities due to land-use change (Tin et al., 2018).
Transformation of tropical forest to, for example OP plantations, reduces bacterial abundance initially,
alters the community composition but once established may not necessarily result in less bacterial richness
in the OP soil (Lee-Cruz et al., 2013; Tripathi et al., 2016). Agricultural soils (including OP soils) are
often thought to promote diversity through management, such as fertilisation and crop inputs and thereby
reduce competition amongst soil microorganisms (Lee-Cruz et al., 2013). Information on microbial
communities will help to understand the impact of anthropogenic land-use change and its impact on
biogeochemical processes (Tin et al., 2018). The lack of our current understanding restricts our ability to
predict and model responses to environmental change (Lee-Cruz et al., 2013). This is particularly





important as 80-90% of soil processes are mediated by microorganisms (Nannipieri et al., 2003). In our
study, we aim to understand whether differences in microbial communities could also help understand
measured differences in greenhouse gas (GHG) emissions. One part of this present study has investigated
potential controlling factors and microbial pathways leading to GHG emissions from soil in controlled
laboratory incubations, which complement the findings presented here from actual field measurements as
the soil was taken from a subset of the sites (Drewer et al., 2020).
The objectives of this study were:

1)  to compare GHG emission rates from different land-uses

2)  to investigate whether management practices and land-use will have a larger effect on GHG fluxes

than other measured abiotic and biotic parameters

3)  to broadly upscale our measurements to Sabah scale


In light of countries committing to reduce and mitigate GHG emissions, e.g. 2015 Paris Agreement
(UNFCCC, 2015), it is important to constrain each country's current emission rates, by providing data
from measurements rather than relying on model estimates. In this study, we present much needed data
of $N_2O$ and $CH_4$ emission rates from logged tropical forests and OP plantations on mineral soil as well as
their biochemical characteristics and temporal and spatial variability. We present two years of
measurements from logged forests and OP plantations in Malaysian Borneo, Sabah from the same
geographical area and on mineral soil.





## 2 Methods

### 2.1 Site description

The present study was carried out within the Stability of Altered Forest Ecosystems (SAFE) project in
Malaysian Borneo (4°49'N, 116°54'E) in 2015 and 2016. The SAFE project was set up in Sabah in 2011
in a secondary forest, designated by the Sabah government for conversion to OP plantations. SAFE is a
long-term landscape-scale experiment designed to study the effects of anthropogenic activities related to
deforestation and OP agriculture on the ecosystem as a whole (Ewers et al., 2011). The main aim of the
SAFE project is to study how habitat fragmentation affects the forest ecosystem, mainly its biodiversity.
The design comprises forest fragments of 1 ha, 10 ha and 100 ha. Larger areas of forests, designated as
continuous logged forests, and not part of the conversion plan, were selected as controls. All forest sites
had been selectively logged for dipterocarps, first in the 1970s then again between 2000 and 2008, such
that the logged forest and forest fragments have a similar land-use history (Ewers et al., 2011). We had
the opportunity to investigate GHG fluxes within this experimental site. As our sampling took place when
conversion was still ongoing (i.e. designated 'fragments' were not fragmented yet), we classify sampling
locations in 'fragments' and 'logged forest' controls both as 'logged forest'. We selected a young OP
plantation, around 2 years old at the time we started measurements (OP2) and a medium aged OP
plantation, around 7 years old at the start of the project (OP7). The riparian area (RR) is adjacent and
down slope from OP7. In addition, we selected a slightly older plantation, around 12 years of age at the
start of the project (OP12). All OP plantations in this study were terraced. Logged forest sites are the 10
ha plots of the logged forest (and future fragments) LF, B and E of the SAFE design.



The climate in the study area is wet tropical with a wet season typically from October to February and a
dry season typically from March to September with average monthly temperatures of 32.5°C (irrespective
of season) and average monthly rainfall of 164.1 mm (climate-data.org, 2019). At SAFE, the mean
monthly rainfall over the two years of study period (2015 and 2016) was 190 mm, ranging from 45 mm
during the driest month (Mar 2015) to 470 mm during the wettest month (Sep 2016; R. Walsh, Figure 1).
Annual rainfall was 1927 mm in 2015 and 2644 mm in 2016 with 2015 being drier than usual. The soils
at SAFE are classed as orthic Acrisols or Ultisols (Riutta et al., 2018).

## 2.2. Field measurements

In order to measure fluxes of $N_2O$ and $CH_4$ from the chosen logged forests and OP plantations, a total of
56 static chambers were installed in the SAFE area. Four chambers were placed in each of the two 10 ha
plots in LF, B, and E, resulting in 8 chambers per site. In the OP plantations, 12 chambers were installed
in the ~7-year old OP plantation, 8 in a ~2-year old, and 8 in a ~12-year old OP plantation. These were
the plantation ages when we started sampling in 2015, hence, the sites are labelled OP2, OP7 and OP12.
For exact GPS locations see the published dataset (Drewer et al., 2019). Fluxes were measured from all
56 chambers every two months over a two-year period, from January 2015 to November 2016; resulting
in 12 measurement occasions for each of the chambers and a total of 672 individual flux measurements.

We received basic fertiliser information from the estate managers at the beginning of our study. Our
measurement sites OP2 and OP7 were managed by the same estate. Fertiliser was applied as slow release





(over 4 – 6 months) bags (500 g) of the brand 'PlantSafe®' (N as Ammonium Sulphate). For palms 0 – 5
years of age PlantSafe® 12-8-16-1.5+trace elements (Diammonium Phosphat ($(NH_4)_2PO_4$), Murite of
Potash (KCl), Ammonium Sulphate ($(NH_4)_2SO_4$), Magnesium Sulphate ($MgSO_4$) + Borax Penthydrate)
was used, and for palms >5 years PlantSafe® 8-8-27-15 was applied as 2 kg bag per plant, three times
per year. Planting density was approximately 9 x 9 m spacing between palms and in addition to the mineral
fertiliser, empty fruit bunches (EFB) were spread, however, there appeared to be no obvious pattern of
application and most EFB were piled up along the main roads, rather than distributed evenly throughout
the plantations. The site OP12 was managed by a different estate. Distance between the palms and
planting density here was 8 x 8 m. Application of fertiliser also occurred as PlantSafe® bags with two
applications a year with 3-4 kg per palm each time, totalling about 8 kg N $ha^{-1}$ $y^{-1}$. EFB were not returned
to this plantation and Glyphosate was applied three times per year around each palm stem to control
weeds. We assume Glyphosate was also applied to the OP2 and OP7 plantations in the other estate.
Generally, fertiliser management was according to recommendations by the Malaysian Palm Oil Board
(MPOB). As our sampling frequency was every two months, we were not able to capture individual
fertilisation events and that was not the scope of this study.

**2.2.1 Soil nitrous oxide ($N_2O$) and methane ($CH_4$) fluxes**
The static chamber method was used for $N_2O$ and $CH_4$ flux measurements as described in previous studies
(Drewer et al., 2017a;Drewer et al., 2017b). Round static chambers (diameter = 40 cm) consisting of
opaque polypropylene bases of 10 cm height were inserted into the ground to a depth of approximately 5





cm for the entire study period. Lids of 25 cm height were fastened onto the bases using four strong clips,
only during the 45-minute measurement periods. A strip of commercially available draft excluder glued
onto the flange of the lid provided a gas tight seal between chamber and lid. The lids were fitted with a
pressure compensation plug to maintain ambient pressure in the chambers during and after sample
removal. Gas samples were taken at regular intervals (0, 15, 30, 45 min) from each chamber. A three-way
tap was used for gas sample removal using a 100 ml syringe. 20 ml glass vials were filled with a double
needle system to flush the vials with five times their volume and remained at ambient pressure rather than
being over-pressurised. The sample vials were sent to CEH Edinburgh for analysis usually between 4-7
weeks after sampling. A specifically conducted storage test confirmed no significant loss of concentration
during that time period. Samples and three sets of four certified standard concentrations ($N_2O$, $CH_4$ in $N_2$
with 20% $O_2$) were analysed using a gas chromatograph (Agilent GC7890B with headspace autosampler
7697A; Agilent, Santa Clara, California) with micro electron capture detector (μECD) for $N_2O$ analysis
and flame ionization detector (FID) for $CH_4$ analysis. These detectors were setup in parallel allowing the
analysis of the two GHGs at the same time. Limit of detection was 5 ppb for $N_2O$ and 40 ppb for $CH_4$.
Peak integration was carried out with OpenLab© Software Suite (Agilent, Santa Clara, California).

The flux F ($\mu g\ m^{-2}\ s^{-1}$) for each sequence of gas samples from the different chambers was calculated
according to Equation 1:
$$F = \frac{dC}{dt} \times \frac{\partial V}{A} \qquad \text{(Equation 1)}$$



Where dC/dt is the concentration (C, µmol mol$^{-1}$) change over time (t, in s), which was calculated by
linear regression, ρV/A is the number of molecules in the enclosure volume to ground surface ratio, where
ρ is the density of air (mol m$^{-3}$), V (m$^3$) is the air volume in the chamber and A (m$^2$) is the surface area in
the chamber (Levy et al., 2012).

Applying the analytical limit of detection to the flux calculation, the resulting detection limits and
therefore uncertainties associated with the flux measurements are 1.6 µg N m$^{-2}$ h$^{-1}$ for $N_2O$ and 5 µg C
m$^{-2}$ h$^{-1}$ for $CH_4$ in the units used in the results section.

**2.2.2 Soil respiration ($CO_2$) fluxes**
In addition, soil $CO_2$ respiration rates were measured close to each chamber location using a dynamic
chamber (volume: 0.001171 m$^3$) covering 0.0078 m$^2$ of soil for 120 s with an EGM-4 infrared gas analyser
(IRGA: InfraRed Gas Analyser; PP Systems; Hitchin, Hertfordshire, England). To do so, cut drainpipes
of 7 cm height matching the diameter of the IRGA chamber were inserted into the ground to a depth of
about 5 cm for the duration of the study to allow for a good seal with the soil surface. All vegetation and
litter was removed from the surface to guarantee soil-only respiration measurements. Taking into account
the time of measurement and the soil temperature, fluxes were calculated based on the linear increase of
$CO_2$ concentrations. Soil respiration was measured every time the static chambers were measured,
resulting in 12 measurement occasions for each of the 56 locations and 672 individual measurements.





### 2.2.3 Auxiliary physical and chemical soil measurements

Other environmental parameters were measured during time of chamber enclosure as possible explanatory variables for correlation with recorded GHG fluxes. Soil and air temperatures were measured using a handheld Omega HH370 temperature probe (Omega Engineering UK Ltd., Manchester, UK) at each chamber location at a soil depth of 10 cm and by holding the temperature sensor 30 cm above the soil surface at chamber height. Volumetric soil moisture content (VMC) was measured at a depth of 7 cm using with a portable probe (Hydrosense 2; Campbell Scientific, Loughborough, UK). For determining KCl-extractable soil nitrogen (N) in the field, soil samples were collected to a depth of 10 cm around each of the chamber locations on each of the chamber measurement days, using a gouge auger. Extractions were carried out in the field laboratory on the same day. Soil samples were mixed well, stones were removed, and subsamples of ca. 6 g soil (fresh weight) was transferred into 50 ml falcon tubes containing 25-ml 1 M KCl solution. The samples were shaken for 1 min every 15 min for one hour, then filtered through Whatman 42© filter paper (GE Healthcare, Chicago, USA) and kept in the fridge after addition of a drop of 75% $H_2SO_4$ as a preservative. Analysis for ammonium ($NH_4^+$) and nitrate ($NO_3^-$) concentrations was carried out at Forest Research Centre in Sandakan (Sabah, Malaysia) using a colorimetric method (Astoria 2 Analyzer (Astoria-Pacific Inc., USA).

The following parameters were measured less frequently. Soil pH was measured on three occasions from the top 0-10 cm, close to each chamber at the start of the measurement period and two months later, and inside the chambers after the last flux measurements at the end of the experiment. For pH measurements



10 g of fresh soil was mixed with deionised $H_2O$ (ratio 1:2), and after 1 hour analysed on a MP 220 pH
meter (Mettler Toledo GmbH, Schwerzenbach, Switzerland). Soil samples for bulk density were collected
from inside each chamber after the final flux measurement. Galvanised iron rings (98.17 $cm^3$) with a sharp
edge were inserted in the upper soil layer with a hammer to 5 cm depth without compaction. Samples
were oven-dried at 105°C until constant weight (usually 48 hours) and bulk density (g $cm^{-3}$) was
calculated based on the dry weight occupying the volume of the ring. Total C and N in soil and litter was
measured once on the last sampling occasion. Soil samples were taken from the top 0-10 cm inside the
chambers. The samples were air dried in the field laboratory and a subsample of each were dried at 105°C
to constant weight in the laboratory to convert the results to oven-dry weight, ground and analysed at the
Forest Research Centre in Sandakan on an elemental analyser (Vario Max CN Elemental Analyzer
(Elementar Analysensysteme, Germany). Litter was collected from the surface area of each chamber, air
dried at 30 °C and analysed for total C and N as described above.

**2.2.4 Soil microbial community composition**
Soil samples for microbial analysis were taken on two occasions from all 56 flux chamber locations. Soil
samples were taken in March 2016 and November 2016 (the last sampling occasion). On the first sampling
date, soil was taken close to each chamber in order not to disturb the soil inside the chamber. In November
2016, soil was taken from inside each chamber, as this was the experimental end date. Approximately 5
g of soil was taken from the top 3 cm and stored in ziplok bags at ambient air temperature until posting
to CEH Wallingford for analysis. The soil samples had to be sent as 'fresh' samples as there were no



freezers operating continuously at the field station, therefore it was not possible to keep the soil frozen
during storage and transport. The samples were frozen at -80°C once they reached CEH Wallingford until
analyses.

For sequencing analyses of bacterial, and fungal and soil eukaryotic communities, DNA was extracted
from 0.2 g of soil using the PowerSoil-htp 96 Well DNA Isolation kit (Qiagen Ltd, Manchester, UK)
according to manufacturer's protocols. The dual indexing protocol of Kozich et al. (2013) was used for
Illumina MiSeq sequencing (Kozich et al., 2013) with each primer consisting of the appropriate Illumina
adapter, 8-nt index sequence, a 10-nt pad sequence, a 2-nt linker and the amplicon specific primer. The
V3–V4 hypervariable regions of the bacterial 16S rRNA gene were amplified using primers 341F
(Muyzer et al., 1993) and 806R (Yu et al., 2005), CCTACGGGAGGCAGCAG and
GCTATTGGAGCTGGAATTAC respectively; the ITS2 region for fungi using primer ITS7f
(GTGARTCATCGAATCTTTG) and ITS4r (TCCTCCGCTTATTGATATGC) (Ihrmark et al., 2012) for
eukaryotes the 18S rRNA amplicon primers from (Baldwin; A.J et al., 2005) were used
(AACCTGGTTGATCCTGCCAGT and GCTATTGGAGCTGGAATTAC). After an initial denaturation
at 95 ℃ for 2 minutes PCR conditions were: denaturation at 95 ℃ for 15 seconds; annealing at
temperatures 55 ℃, 52 ℃, 57 ℃ for 16S, ITS and 18S reactions respectively; annealing times were 30
seconds with extension at 72 ℃ for 30 seconds; cycle numbers were 30; final extension of 10 minutes at
72 ℃ was included. Amplicon concentrations were normalized using SequalPrep Normalization Plate Kit
(Thermo Fisher Scientific Ltd, Altrincham, UK) prior to sequencing each amplicon library separately on
the Illumina MiSeq using V3 chemistry using V3 600 cycle reagents at concentrations of 8 pM with a 5%
PhiX Illumina control library (Illumina Ltd, Cambridge, UK).

Illumina demultiplexed sequences were processed in R software package, version 3.6.1 (R Core Team,
2017) using DADA2 (Callahan et al., 2016) to quality filter, merge, denoise and construct sequence tables
as follows:  Amplicons reads were trimmed to 270 and 220 bases, forward and reverse respectively for
ITS, and forward reads were trimmed to 250 and 280 bases for 16S and 18S respectively. Filtering settings
were maximum number of Ns (maxN) = 0, maximum number of expected errors (maxEE) = (1,1).
Sequences were dereplicated and the DADA2 core sequence variant inference algorithms applied.
Forward and reverse reads were merged using mergePairs function as appropriate. Sequence tables were
constructed from the resultant actual sequence variants and chimeric sequences were removed using
removeBimeraDenovo default settings.

**2.3 Data analysis**
Environmental data, especially soil $N_2O$ fluxes, are typically highly variable in space and time, which
makes their analysis challenging. Much of the variation cannot be explained by co-variates, as the driving
microbial processes are not directly observed. They are also usually strongly left skewed (containing a
high number of very small fluxes), and are expected to approximate a lognormal distribution. Against this
background, trying to detect effects of land-use (or experimental treatments) is difficult. The calculation



of a confidence interval on the mean of a log-normal distribution is problematic when variability is high
and sample size is small (e.g. Finney 1941), as is generally the case with flux measurements.

Here we applied a Bayesian methodology to address this problem, using a model similar to that described
by Levy et al. (2017). This accounts for the lognormal distribution of observations, while including
hierarchical effects of land-use, and effects of sites within land-use types as well as the repeated measures.
In the current statistical terminology, this is a generalised linear mixed-effect model (GLMM) with a
lognormal response and identity link function. The model consists of a fixed effect of land-use (Forest,
Oil Palm, or Riparian), with a random effect representing the variation among sites within a land-use type.
The parameters were estimated by the Markov chain Monte Carlo (MCMC) method, using Gibbs
sampling as implemented in Just Another Gibbs Sampler (JAGS) (Plummer 1994), and described in more
detail by Levy et al. (2017).

All other statistical analyses were conducted using the R software package, version 3.4.3 (R Core Team,
2017) using the lme4 package for linear mixed-effects models (Bates et al., 2015) and ordinary multiple
regression. Model selection was examined by sequentially dropping terms and assessing AIC and similar
criteria using the MuMIn package (Bartoń, 2013). For $N_2O$ and $CH_4$, where negative values occurred, the
minimum was added to all data points (-30 and -115 µg $m^{-2}$ $h^{-1}$, respectively) so that a lognormal
distribution could be fitted.





For microbial community composition samples within each sampling point were assessed in R for
sequencing depth. Samples with fewer than 4000 reads were deemed as containing insufficient data and
discarded. Package Vegan was used to rarefy each sampling occasion's samples to the minimum read
number. Vegan functions specnumber, diversity and metaMDS were used to generate the statistics for
richness, Shannon's diversity and Nonmetric Multidimensional Scaling, respectively. Analysis of
similarities (ANOSIM) was used to test statistically whether there was a significant difference between
two or more groups of parameters in relation to the microbial communities.

**2.4 Upscaling of N$_2$O fluxes to Sabah scale**
In an attempt to broadly upscale our findings, we calculated the annual soil N$_2$O emission for the Sabah
state based on the data from this study (Table 2), together with land cover areas estimates (Gaveau et al.,
2016) of forests, pulpwood and OP plantations for 1973 and six 5 yearly intervals from 1990-2015. We
included the pulpwood plantation area in the total forest area, as to our knowledge there are no data of
N$_2$O emissions from this sector. We used mean emissions and the 95% confidence interval calculated by
the GLMM and posterior probability to account for variability and associated uncertainties.
**3 Results**
**3.1 Soil parameters**
Results are presented by site (B, E, LF, OP2, OP7, OP12, RR) or land-use (logged forest (B, E, LF), oil
palm (OP2, OP7, OP12), riparian (RR)). Soil pH was acidic from logged forest site B (pH 3.65±0.44)





compared to forest E and LF, which were closer to neutral (pH 6.38±0.67 and 6.14±0.5), and the OP
plantations were more acidic (pH 4.5-4.7±0.2) compared to the riparian area (pH 5.8±0.55) (Table 1).
Bulk density was lower at the forest sites (~0.81 g cm$^{-3}$ ) compared to the OP plantations (~1.26 g cm$^{-3}$)
mainly due to a higher amount or organic matter and litter in the forest sites (B, E, LF) and a combination
of compaction due to land management and lower organic matter content in the OP plantations and
riparian area (OP2, OP7, OP12, RR) (Table 1).  Total carbon (C) and nitrogen (N) in soil were higher in
the logged forest sites (~3-7% C and ~0.25-0.4% N, albeit with a very high variability) than the OP
plantations  (<1% C and <0.1% N) (Table 1) due to larger amount of litter present. The riparian reserve
had higher content of C and N in the soil (1.2% C, 0.15% N) than the OP plantations but not as high as
the logged forests. Variability even within one site was large for the forest sites which is also reflected in
the C/N ratios (Table 1). Litter was present in all of the forest and riparian reserve chambers and only in
a few of the OP chambers. The average litter weight in the forest chambers was between 50 and 150 g
dry weight with a very high variability, about 15 g in the riparian area, and hardly any litter in the OP
chambers, with no litter in OP12, only in one of the OP7 chambers and an average amount of 50 g of litter
in the young OP2 , again with a very high variability (Table 1). The total C and N content in litter was
similar in logged forest and OP (~35-40% C and ~1.5-1.8% N); the main difference was the
presence/absence of litter and the amount present. For all these measured parameters the variability within
each site was high apart from pH in OP which was most likely regulated by plantation management
operations. None of the soil physicochemcial parameters were significantly different for the different
land-uses or sites apart from pH from site B.






Soil moisture had high variability both spatially and temporaly, with a large range for all land-uses (Figure
2a) and no discernable temporal trend. The riparian reserve tended to have slightly higher soil moisture
than the adjacent OP plantation due to proximity to a little stream and ground cover vegetation. The
highest soil temperatures were measured in the young OP which had no canopy closure or shaded areas
(Figure 2b). Soil temperature was slightly higher in the riparian reserve than the adjacent OP7, likely due
to softwood trees with much less canopy cover compared to the 7 year old OP plantation. In summary,
there was no discernible temporal trend of soil moisture or temperature over the two year measurement
period and no apparent difference between wet and dry seasons.

Soil extractable mineral N (both $NH_4^+$ and $NO_3^-$) was highly variable across the OP plantations with mean
values of 8±23 and 6.3±18 mg N $g^{-1}$, respectively, 4.5±5 and 2.3±4 mg N $g^{-1}$ in riparian and 3.9±5 and
5.3±5 mg N $g^{-1}$ in the forests (Figure 3, Table 2). We measured the lowest average $NH_4^+$ and $NO_3^-$
concentrations in the 12 year old plantation (OP12), and the highest in the youngest OP plantation (OP2)
with maxima of >150 mg $g^{-1}$, however, with a very high spatial variability (Figure 3, Table 2). Due to the
low frequency of soil and flux sampling (every 2 months), and the lack of knowldege of the fertilisation
dates, it is not possible to correlate soil mineral N concentrations with  individual fertiliser events. $NH_4^+$
and $NO_3^-$ concentrations of  the logged forest sites, older OP plantation and riparian reserve were very
similar.



### 3.2 Greenhouse gases

### 3.2.1 Nitrous oxide (N₂O)

There were no temporal trends of nitrous oxide (N$_2$O-N) fluxes and no distinct differences between wet (usually Oct to Feb) and dry (Mar to Sep) seasons (Figure 4a). Variability in N$_2$O-N fluxes for all sites was high and the largest range was measured in the OP plantations (Figure 4a, Table 2). We find that the largest fluxes observed were from the young (OP2) and old (OP12) oil palm plantations and exceed 1500 µg m$^{-2}$ h$^{-1}$ N$_2$O-N for individual chambers. In the logged forest, largest fluxes were ~400 µg m$^{-2}$ h$^{-1}$ for individual chambers at site B. On a given day, very large as well as very small fluxes were measured in the OP plantations. For each land-use standard deviation was a lot larger than the mean (Table 2); logged forest 13.9±171 µg m$^{-2}$ h$^{-1}$ N$_2$O-N, OP 46.2±166 µg m$^{-2}$ h$^{-1}$ N$_2$O-N and riparian 31.8±220 µg m$^{-2}$ h$^{-1}$ N$_2$O-N. By fitting the GLMM to the data, we estimated the posterior probability density of the effect of land-use on N$_2$O flux: mean fluxes to be 13.9 (95 % CI: -6.3 to 41.5) µg m$^{-2}$ h$^{-1}$ for logged forests, 46.2 (18.4 to 97.5) µg m$^{-2}$ h$^{-1}$ for OP and 31.8 (-6.3 to 130.0) µg m$^{-2}$ h$^{-1}$ for the riparian area (Figure 4b, Table 2). The output using the Bayesian approach can be interpreted as follows: The area of the OP curve does not overlap with the area of the forest curve, which means that the probability is higher that the flux from OP plantation is higher than the flux from logged forest, with the riparian zone being intermediate. To investigate effects of additional variables, we used the automated model selection algorithm in the MuMIn R package, which uses all possible combinations of fixed effect terms and ranks them by AIC (Bartoń, 2013). Possible terms included land-use, pH, soil moisture, NH$_4^+$, NO$_3^-$, bulk density, soil and air temperature, and the microbial NMDS axes. This procedure found the inclusion of NH$_4^+$ and NO$_3^-$, soil





moisture and soil temperature, in addition to land-use, to give the optimal model. However, whilst land-
use (including the site-level effects) explained 13% of the variance (expressed as conditional $R^2$, (Bartoń,
2013)), the additional four terms increased this by only 4%. The microbial NMDS axes did not improve
the model fit, as measured by AIC.

**3.2.2 Methane (CH₄)**
For methane, both negative fluxes (= $CH_4$ oxidation) and positive fluxes ($CH_4$ emission) were measured
at all sites throughout the measurement period (Figure 5).  Highest emission and uptake rates were
measured in the logged forest sites, with emissions reaching almost 300 µg m$^{-2}$ h$^{-1}$ $CH_4$-C at site E, and
uptake rates of up to 85 µg m$^{-2}$ h$^{-1}$ $CH_4$-C at sites LF and B. In the OP plantations highest emissions were
measured at OP7 (~100 µg m$^{-2}$ h$^{-1}$ $CH_4$-C), and uptake rates were <50 µg m$^{-2}$ h$^{-1}$ $CH_4$-C. Overall, $CH_4$
flux ranges were larger in the logged forests than OP plantations. Grouping fluxes by land-use, mean
fluxes were about 2.2±48.3 µg $CH_4$-C m$^{-2}$ h$^{-1}$ for logged forest, -2.6±17.2 µg $CH_4$-C m$^{-2}$ h$^{-1}$ for OP and
1.3±12.6 µg $CH_4$-C m$^{-2}$ h$^{-1}$ for riparian reserve (Table 2). The magnitudes of $CH_4$-C fluxes in the riparian
reserve were more similar to the logged forests sites than the OP plantations. Standard deviation again
was large but not as large as for $N_2O$.

As for $N_2O$, possible drivers of $CH_4$ fluxes were investigated using linear mixed effect models and the
same model selection methods. However, no correlations with co-variates could be established, even with
land-use. For example, a model including terms for land-use, pH, soil moisture, $NO_3$, $NH_4$, bulk density,





soil and air temperature could explain only 3% of the variance. Land-use was clearly not a strong
determinant of $CH_4$ flux, and the posterior distributions are not shown.

**3.2.3 Soil respiration ($CO_2$)**
Soil respiration $CO_2$-C fluxes also had a high spatial variability (Figure 6). There was a trend to slightly
higher respiration rates at logged forest sites than OP plantations. Grouping fluxes by land-use, gave mean
fluxes of $137.4\pm95$ mg m$^{-2}$ h$^{-1}$ for logged forests, $93.3\pm70$ mg m$^{-2}$ h$^{-1}$ for OP plantations and $157.7\pm106$
mg m$^{-2}$ h$^{-1}$ for the riparian site (Table 2). Soil respiration in the measured riparian reserves was therefore
in the range of the soil respiration of logged forest, which was higher than from OP sites. Data was log
transformed before statistical analysis. A linear mixed-effects model including all terms could explain
25% of the variance, and land-use alone explained 7% of the variance.

**3.3 Soil biodiversity**
Soil samples for biodiversity measurements were collected in the low rainfall month, March 2016 (~50
mm), and the high rainfall month, November 2016 (~250 mm, Figure 1), in order to quantify broad
differences in communities due to land-use and provide additional biodiversity variables for modelling
fluxes. Three different amplicon sequencing assays were performed on extracted DNA targeting bacteria
(16S rRNA gene), fungi (ITS region), and broad groups of soil eukaryotic taxa (18S rRNA gene, including
principally fungi, protists and algae). The ordinations and multivariate permutation effects of land-use
were generally consistent across the two sampling points irrespective of seasonal climatic differences





(Figure 7). Fitting environmental vectors to the ordination axis scores revealed that the bacterial
communities were highly related to soil pH ($r^2$ = 0.85 and 0.84, p<0.001, for the two sample dates
respectively), with acid soils (pH 3.6) at site B, compared to near neutral pH of 6.1 and 6.4 at sites LF
and E, Table 1).    Weaker relationships with the land-use factors ($r^2$= 0.23 and 0.11, p<0.05) were
observed. Logged forests E and LF had very similar bacterial communities, which were distinct from the
three OP sites and also the riparian site.    In contrast, fungal and eukaryotic communities were not as
strongly related to soil pH (fungal $r^2$= 0.67 and 0.72, and eukaryotic $r^2$ = 0.73 and 0.79 for the two sample
dates respectively, p<0.001), and were more strongly related to above ground land-use than bacterial
communities (fungal $r^2$= 0.52 and 0.57, and eukaryotic $r^2$ = 0.50 and 0.42, p<0.001). As can be seen in
the fungal ordinations particularly, the forested sites formed a distinct cluster separate from the OP sites,
despite the large differences in soil acidity.

**3.4 Upscaling of N$_2$O fluxes to Sabah scale**

In an attempt to broadly upscale our findings, we calculated the annual soil N$_2$O emission for the Sabah
state based on the data from this study (Table 2), together with land cover areas estimates (Gaveau et al.,
2016). Nitrous oxide emissions calculated for the Sabah region showed a strong dependence on the
conversion of forest to OP plantations from 1973 to present day. By 2015, the total estimated N$_2$O
emissions from OP plantations were roughly 40% of total emissions, with 60% of the emissions from
forested areas, despite the OP area being less than 40% of the forest area. The Sabah scale median N$_2$O
emission estimate had increased from 7.6 Mt (95% confidence interval, -3.0-22.3 Mt) per year in 1973 to





11.4 Mt (0.2-28.6 Mt) per year in 2015. As the measured $CH_4$ fluxes were fluctuating around zero, the
changes in land-use also resulted in small changes of $CH_4$ flux rates over the 42-year period. Our median
results suggest that Sabah is a sink for $CH_4$ (4 Mt $y^{-1}$) throughout the time period presented.

**4 Discussion**

This study focussed on comparing GHG fluxes from different land-use types in the Tropics. Our data,
although not high frequency measurements, provide a comprehensive insight in the potential impact of
converting logged forests to OP plantations on GHG fluxes. The focus of this study is on $N_2O$, with
auxiliary measurements of $CH_4$ and soil respiration. To date only four studies published data of $N_2O$
emissions from OP plantations on mineral soil in Southeast Asia using the chamber method that included
measurements from a time period of longer than 6 months (Skiba et al., 2020). Only one of these studies
included measurements in Malaysia (Sakata et al., 2015). Globally tropical forests are the largest natural
source of $N_2O$ (Werner et al., 2007). Therefore, the question is whether the N input to OP plantations with
lower organic matter (TC/TN content) compared to tropical forests (lots of organic matter input, warm,
humid), lead to larger $N_2O$ emissions than forest. Although it has been recognised that $N_2O$ emissions are
induced by N-fertiliser application in OP, when considering annual or long-term emissions from mineral
soil, these fertilisation patterns might not have a pronounced or clear effect (Kaupper et al., 2019). For
example, N-fertiliser induced $N_2O$ fluxes comprised only 6-21% of the annual soil $N_2O$ fluxes in OP
plantations in Sumatra, Indonesia (Hassler et al., 2017), the rest was due to other natural processes
occurring in the soil. Therefore, our study can be considered representative, particularly as measurements



were carried out over two years. All three land-use types (logged forest, oil palm and riparian) showed

positive $N_2O$ fluxes albeit with a high variability.

On some occasions, our measured fluxes exceeded the range reported by Shizuka et al. (2005) of $N_2O$

emissions from OP plantations on mineral soil in Indonesia, ranging from ~1-29 µg m$^{-2}$ h$^{-1}$, by an order

of magnitude (maximum measured 350 µg m$^{-2}$ h$^{-1}$). The highest values reported by Shizuka et al. (2005)

were from young plantations, while the lowest were reported from older plantations. They suggested the

low N uptake of young plantations after fertiliser application and the fixation of N by the legume cover

crop could be the reason for the high emissions. On the other hand, the low emissions from older

plantations could result from higher N uptake by the OP and the absence of legume cover. In their study,

$N_2O$ emissions were mainly determined by soil moisture (Ishizuka et al., 2005); which was not the case

here. Mean $N_2O$ fluxes from a sandy soil in Malaysia were reported to range from 0.80 to 3.81 and 1.63

to 5.34 µg N m$^{-2}$ h$^{-1}$ in the wet and dry seasons, respectively (Sakata et al., 2015). This was lower than

from a sandy loam soil in Indonesia (27.4 to 89.7 and 6.27 to 19.1 µg N m$^{-2}$ h$^{-1}$ in the wet and dry seasons,

respectively) (Sakata et al., 2015). Despite the limited number of measurements in OP plantations on

mineral soils and the high variability of results, emissions seem to generally be higher in the early years

of the OP plantations (Pardon et al., 2016a). This is not necessarily reflected in our data, as the OP2

(young) and OP12 sites (older) showed higher fluxes than the OP7 (medium age) site; though with a

lifespan of up to 30 years, all plantations measured in this study can still be regarded as immature. As in

our study, Aini et al. (2015) also found no differences in $N_2O$ fluxes in the wet and dry months with fluxes





ranging from 0.08 to 53 µg N m$^{-2}$ h$^{-1}$. The range of our measured fluxes exceeded those of these previously
published studies. However, it is difficult to generalise, as variability appeared to be high in all studies.

Our measured N$_2$O fluxes from the riparian area were similar to those measured in the OP plantation, as
soil properties were more similar to OP than logged forest. There is currently a knowledge gap on GHG
emissions from riparian buffers (Luke et al., 2019) and more studies are needed to evaluate the
effectiveness in terms of nutrient retention and potential GHG mitigation of such buffers. A previously
published study from Peninsula Malaysia reported mean N$_2$O emission rates from logged tropical forest
sites ranging from 17.7 to 92.0 µg m$^{-2}$ h$^{-1}$ N$_2$O-N which was significantly larger than from their measured
unlogged sites (Yashiro et al., 2008). Even though the range of our measured fluxes from logged forest
sites was wider, it is broadly in the same order of magnitude (13.9±171 µg m$^{-2}$ h$^{-1}$ N$_2$O-N).

As often the case with GHG studies, the variation in the measured GHG fluxes could not be explained
with certainty by any of the measured soil parameters. Our sampling frequency was not high enough to
investigate, for example, emission rates after fertiliser application in the OP plantations and besides, this
was not the aim of our study. The wide ranges we measured for soil mineral N concentrations and N$_2$O
fluxes were likely due to the spatial and temporal variability of the fertiliser application, as the slow
release fertiliser bags were randomly placed around the trees, and with time, the fertiliser release rate
slowed down. Apart from no strong correlations with single environmental factors, multiple regression
and mixed models were only able to explain around 17% of the variance including multiple measured



parameters. However, applying the Bayesian method, the posterior probability density of the effect of
land-use on N$_2$O flux confirmed that fluxes from the OP plantations were evidently higher than those
from the forests (the area of the OP curve does not overlap with the forest curve), with the riparian zone
being intermediate (mean fluxes 13.9 (95 % CI: -6.3 to 41.5) µg m$^{-2}$ h$^{-1}$ for logged forests, 46.2 (18.4 392
to 97.5) µg m$^{-2}$ h$^{-1}$ for OP and 31.8 (-6.3 to 130.0) µg m$^{-2}$ h$^{-1}$ for the riparian area).

Agricultural soils such as OP soils can be methane sinks, with uptake rates usually being lower than in
forest soils (Hassler et al., 2015) which could also be seen in our data with logged forest showing higher
uptake rates but at the same time also showing the highest emission rates. However, we did not see the
seasonal cycle reported in Hassler et al., (2015) from Indonesia and generally differences between all
three land-uses (logged forest, oil palm and riparian) were small. The lack of seasonal variability seen in
our study might be due to the fact that dry and wet seasons are not as pronounced in Sabah as in other
tropical regions (Kerdraon et al., 2020) and that temperature is fairly constant throughout the year.

Higher soil respiration (sum of heterotrophic and autotrophic respiration) is often considered as a sign of
good soil health, it reflects the capacity of soil to support soil life including microorganisms and crops.
Heterotrophic soil respiration defines the level of microbial activity, soil organic matter content and its
decomposition whilst autotrophic respiration is the metabolism of organic matter by plants. In a recently
published study investigating litter decomposition, soil respiration fluxes in Sabah (also in the SAFE area)
were higher from forest than OP (Kerdraon et al., 2020). This was also the general trend in our study





despite the high variability of all measured fluxes. Litter input in our plots was larger in the logged forest
plots and riparian reserve than the OP. In litter decomposition experiments, in both Borneo and Panama,
litter input was more important than litter type, which stresses the importance of the amount of
aboveground litter for soil processes in general, especially in disturbed habitats or forest converted to
plantations (Kerdraon et al., 2020).

Analyses of soil microbial communities with different assays targeting different microbial components,
revealed strong influences of soil properties such as pH, but also highlighted that fungal and eukaryotic
communities were more affected by management and land-use than bacteria. Soil pH is known to have
an impact on soil microbial community in the Tropics (Kaupper et al., 2019;Tripathi et al., 2012) which
may explain the very different bacterial communities in logged forest B with the lowest measured pH of
all our sites. Typically, C and N availability or generally soil fertility is known to decrease after
deforestation (Allen et al., 2015; Hassler et al., 2017; Hassler et al., 2015; Kaupper et al., 2019), this is
also reflected in our data (Table 1), especially the very low total N values in all OP plantations. Nutrient
input through litter is higher in the forest than OP plantations and consistently replenished (Guillaume et
al., 2015). Therefore, for microorganisms, OP plantations represent a nutrient deprived environment
(Kaupper et al., 2019). Low total C input can also limit the methanotrophic population size and hence
limit $CH_4$ uptake (Krause et al., 2012). Lower N in OP soil has also shown to limit $CH_4$ uptake when
compared with forest soil (Hassler et al., 2015). Exactly how shifts in C and N after converting forest to
OP may affect processes involved in $N_2O$ and $CH_4$ fluxes remains highly uncertain (Kaupper et al., 2019).



On mineral soil, changes in bulk density after conversion from forest to plantation are often marginal
(Aini et al., 2015; Chiti et al., 2014), however in our study we did see a distinct difference between logged
forest and OP soil (Table 1), which was likely due to the higher organic matter content in the logged forest
soil.

We found distinct differences of microbial communities in the different land-uses. In a recently published
study of a natural rainforest and an OP plantation in Sabah, bacterial community diversity (richness and
evenness) was comparable or even slightly higher in the OP site (Tin et al., 2018). Also, Kaupper et al.
(2019) have  suggested that microbial biodiversity loss occurs soon after clearance and that bacterial
diversity might either be resilient to the change or changes cannot be detected after a sufficient recovery
period (>8 years) after deforestation (Kaupper et al., 2019). Agricultural OP soil has previously been
found to be more functionally diverse compared to forest soil (Mendes et al., 2015; Tripathi et al., 2016)
while microbial functioning in forest soil appears to be dependent on microbial abundance rather than
diversity (Mendes et al., 2015). Reason for this could be that in agricultural soils (i.e. OP plantations)
there is a need for functional diversity in order to maintain a sufficient level of idleness for continued
functioning under stress events such as deforestation and soil management. Despite these few recent
studies on microbial communities, the link to processes leading to GHG fluxes has not been made
(Kaupper et al., 2019), hence predictions on the impact of land-use change are difficult to make. Despite
our data showing land-use and soil property effect on components of the microbial community, inclusion
of derived community metrics in models to predict fluxes did not improve fits; it is possible that a more





specific focus on relevant functional gene abundances will yield greater predictive ability. In a laboratory
incubation study that used soil from some of these field study sites, it was found that both logged forest
and OP soil had the same potential for substantial $N_2O$ fluxes under laboratory conditions (Drewer et al.,
2020). However, under these controlled conditions, riparian reserve soil had negligible $N_2O$ fluxes, which
is in contrast to the fluxes measured in the field. The same study also concluded that despite the high
variability found amongst replicates, the main contribution to $N_2O$ emissions came from proteobacterial
*nirS* and *AniA-nirK* containing denitrifiers and archaeal ammonia oxidizers (Drewer et al., 2020). The
conversion of forest to monoculture plantations is a big threat to ecosystem functioning (Tripathi et al.,
2016), yet we are still missing data on microbial communities to make accurate predictions.

Plantation management, for example returning palm fronds and empty fruit bunches to the plantation soil,
will likely change nutrient cycling (Pardon et al., 2017) and therefore microbial composition. Presence
of, for example, leaf litter as a source of organic matter is essential to maintain soil processes (Kerdraon
et al., 2020). It is vital to understand underlying longer-term processes that ultimately might regulate
GHG fluxes to be able to develop GHG mitigation strategies. More environmentally friendly plantation
management would likely help with maintaining ecosystem functioning and reduce GHG emissions.

In an attempt to broadly upscale our findings, we calculated the annual soil $N_2O$ emission for the Sabah
state based on the data from this study (Table 2), together with land cover areas estimates (Gaveau et al.,
2016). The Sabah scale median $N_2O$ emission estimate had increased from 7.6 Mt per year in 1973 to



11.4 Mt per year in 2015. However, this change is small considering the associated uncertainties,
demonstrated by the interquartile range, -3.0-22.3 Mt per year in 1973 and 0.2-28.6 Mt per year in 2015.
The changes in land-use resulted in small changes of $CH_4$ flux rates over the 42-year period. Our median
results suggest that Sabah is a sink for $CH_4$ (4 Mt $y^{-1}$) throughout the time period presented. There was a
slight decrease to the interquartile range of our estimate as more land was converted to OP plantation,
suggesting that the strength of the sink decreased. However, this is much lower than the uncertainty
associated with this analysis, hence; it is difficult to draw strong conclusions.

## 620 5 Conclusions

$N_2O$ emission rates in Sabah on mineral soil were higher from OP than logged forest over a two-year
study with $N_2O$ emission rates from riparian intermediate. Mean $CH_4$ fluxes were low with very high
variability, showed no clear trend and the highest range of fluxes was measured in logged forests. Fungal
and eukaryotic communities were related to management whilst bacterial communities were strongly
affected by soil pH, which might have masked any management impacts. Mixed models and multiple
regression analysis could only explain 17% of the variation in the measured $N_2O$ fluxes, 3% of the $CH_4$
fluxes and 25% of soil respiration, despite the large number of measured abiotic and biotic parameters.
This is not uncommon for GHG fluxes, but demonstrates that many more studies, ideally at high temporal
and spatial resolution, are required to inform on the impact of land-use and climate change on GHG
fluxes. Scaling up measured $N_2O$ fluxes to Sabah using land areas for forest and OP (Gaveau et al., 2016)
imply that the emissions have increased over the last 42 years, with the proportion of emissions from OP





plantations increasing in comparison to the emissions from forests. Using the range of measured fluxes
with mean and interquartile ranges highlights the large uncertainties still associated with these emission
estimates, despite having almost 700 individual data points over two years. For $CH_4$, the picture is even
more uncertain. More studies on GHG fluxes from tropical forests and OP plantations on mineral soils
(including experiments deriving $N_2O$ emission factors) are needed to reduce the uncertainty of their
emission rates. Furthermore, the impact of current management systems and future potentially more
environmentally friendly plantation management needs to be investigated in order to predict how to
maintain ecosystem function and biodiversity which could have a positive impact on reducing GHG
emissions.



**Data availability**

Drewer, Julia, Leduning, Melissa, Sentian, Justin, & Skiba, Ute. (2019). Soil greenhouse gas fluxes and associated parameters from forest and oil palm in the SAFE landscape [Data set]. Zenodo. http://doi.org/10.5281/zenodo.3258117

**Author contributions**

JD&US designed the project, ML carried out field measurements with help of JD&US and JS as local collaborator. RG&TG carried out microbial analysis. PL carried out statistical analysis. NC assisted with data analysis. ECP&GH carried out upscaling, NM supervised soil parameter analysis. JD wrote the manuscript with contributions from all co-authors.

**Competing interests**

No conflict of interest to declare

**Acknowledgements**

Special thanks to the (LOMBOK) RAs ('Noy' Arnold James, and 'Loly' Lawlina Mansul) at SAFE for help with the field sampling, Fifilyana Abdulkarim for laboratory analysis, and Jake Bicknell for discussions about upscaling. This project was funded as LOMBOK (Land-use Options for Maintaining BiOdiversity and eKosystem functions) by the NERC Human Modified Tropical Forest (HMTF) research programme (NE/K016091/1).





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





**Tables and Figures**

**Table 1.** Soil physicochemical parameters: pH (mean of three sampling occasions and replicate chambers at each site); bulk density (mean of replicate chambers at each site from one sampling occasion); total C and total N in soil from the top 1-10 cm and leaf litter in the chambers (from replicate chambers on one sampling occasion), from the different sites (LF (n=8), B (n=8), E (n=8) = logged forest, OP2 (n=8), OP7 (n=12), OP12 (n=8) = oil palm, RR (n=4) = riparian reserve).

| site | pH | | bulk density [g cm$^{-3}$] | | soil total N [%] | | soil total C [%] | | C/N (soil) | | Total litter dry mass [g] | | litter total N [%] | | litter total C [%] | |
|---|---|---|---|---|---|---|---|---|---|---|---|---|---|---|---|---|
| | mean | sd | mean | sd | mean | sd | mean | sd | mean | sd | mean | sd | mean | sd | mean | sd |
| LF | 6.14 | 0.50 | 0.80 | 0.16 | 0.24 | 0.14 | 3.21 | 2.04 | 14.4 | 4.97 | 53 | 18.18 | 1.76 | 0.39 | 36.44 | 6.82 |
| B | 3.65 | 0.44 | 0.80 | 0.11 | 0.30 | 0.07 | 4.65 | 1.23 | 15.5 | 1.47 | 114 | 51.97 | 1.51 | 0.31 | 33.78 | 7.33 |
| E | 6.38 | 0.67 | 0.84 | 0.21 | 0.38 | 0.26 | 6.40 | 6.72 | 13.8 | 5.44 | 92 | 41.38 | 1.82 | 0.15 | 40.01 | 3.88 |
| OP2 | 4.54 | 0.21 | 1.22 | 0.12 | 0.05 | 0.02 | 0.70 | 0.21 | 14.0 | 1.81 | 53 | 70.54 | 1.78 | 0.28 | 40.62 | 5.88 |
| OP7 | 4.71 | 0.22 | 1.28 | 0.18 | 0.07 | 0.05 | 0.97 | 0.47 | 15.2 | 4.18 | 19[*] | N/A | 1.54 | N/A | 31.99 | N/A |
| OP12 | 4.60 | 0.14 | 1.27 | 0.07 | 0.08 | 0.03 | 0.72 | 0.15 | 9.3 | 2.34 | N/A | N/A | N/A | N/A | N/A | N/A |
| RR | 5.77 | 0.55 | 1.25 | 0.10 | 0.14 | 0.06 | 1.18 | 0.32 | 9.6 | 3.61 | 17 | 3.00 | 1.78 | 0.28 | 40.62 | 5.88 |

[*]only one of the OP7 chambers had litter present



**Table 2.** Greenhouse gas fluxes ($N_2O$-N, $CH_4$-C, soil respiration $CO_2$-C) and soil mineral nitrogen ($NH_4$-
N and $NO_3$-N) averaged over the entire measurement period (January 2015 – November 2016) by land-
use. N = number of individual data points, sd = standard deviation; forest = logged forest, OP = oil palm,
RR = riparian reserve.

| Variable | Land-use | N | Mean | SD | Median |
|---|---|---|---|---|---|
| $N_2O$-N | forest | 286 | 13.87 | 171.49 | 13.90 |
| ($\mu g\ m^{-2}\ h^{-1}$) | OP | 335 | 46.20 | 166.35 | 45.84 |
| | RR | 48 | 31.83 | 220.40 | 30.86 |
| | | | | | |
| $CH_4$-C | forest | 216 | 2.20 | 48.34 | -5.63 |
| ($\mu g\ m^{-2}\ h^{-1}$) | OP | 251 | -2.57 | 17.18 | -3.00 |
| | RR | 36 | 1.27 | 12.60 | -0.38 |
| | | | | | |
| $CO_2$-C | forest | 288 | 137.39 | 94.63 | 115.35 |
| ($mg\ m^{-2}\ h^{-1}$) | OP | 336 | 93.30 | 69.65 | 75.55 |
| | RR | 48 | 157.70 | 105.80 | 142.60 |
| | | | | | |
| $NH_4$-N | forest | 288 | 3.92 | 5.41 | 2.85 |
| $mg\ g^{-1}$ | OP | 336 | 7.99 | 22.72 | 2.50 |
| | RR | 48 | 4.50 | 5.40 | 2.50 |
| | | | | | |
| $NO_3$-N | forest | 288 | 5.30 | 5.28 | 3.40 |
| $mg\ g^{-1}$ | OP | 336 | 6.32 | 18.16 | 1.40 |
| | RR | 48 | 2.25 | 4.19 | 1.35 |



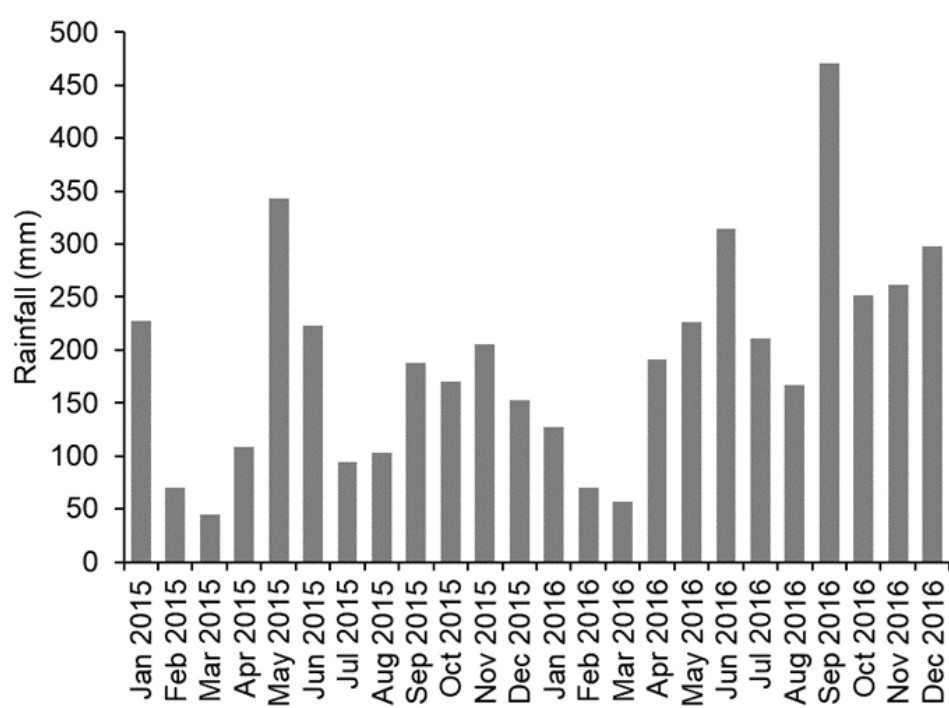


**Figure 1.** Monthly rainfall (mm) in the SAFE area in 2015 and 2016 (R. Walsh).







**Figure 2.** Barplots of mean volumetric soil moisture (a) and mean soil temperature (b) from January 2015

- November 2016, every two months: (upper panel: B, E, LF = logged forests, middle panel: OP12, OP2,

OP7 = oil palm plantations, bottom panel: RR = riparian reserve).


**Figure 3.** Mean mineral N as KCl extractable $NH_4^+$ (a) and $NO_3^-$ (b) from January 2015 - November 2016, every two months (upper panel: B, E, LF = logged forests, middle panel: OP12, OP2, OP7 = oil palm plantations, bottom panel: RR = riparian reserve). Error bars represent standard deviation of the samples around the mean.



**Figure 4. a)** Nitrous oxide ($N_2O$-N) fluxes in µg m$^{-2}$ h$^{-1}$ from January 2015 - November 2016, every two months (upper panel: B, E, LF = logged forests, middle panel: OP12, OP2, OP7 = oil palm plantations, bottom panel: RR = riparian reserve). Bars are mean for each site and error bars are standard deviation of number of chambers per site.


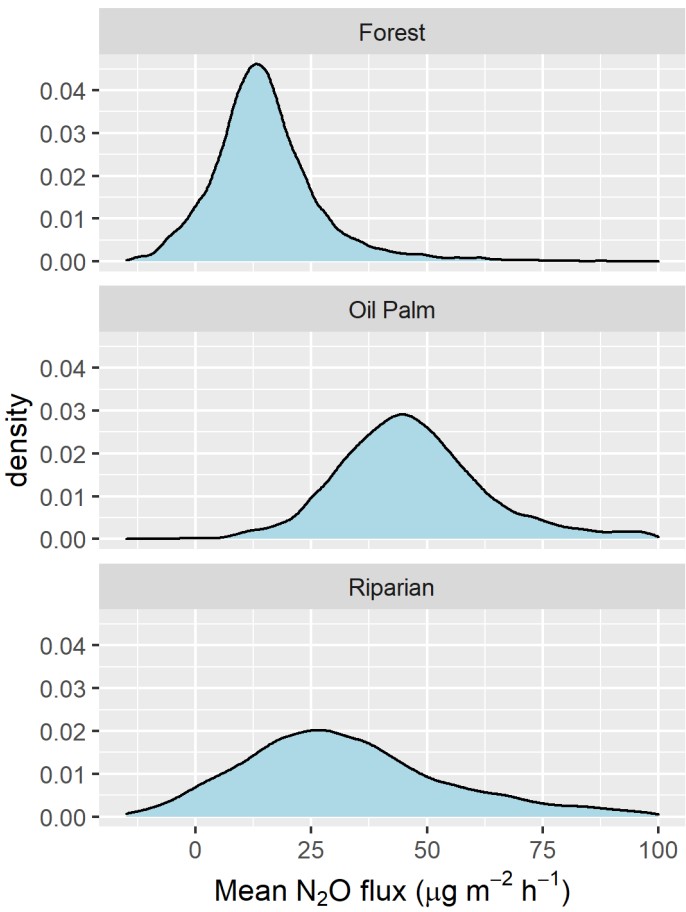

893

**Figure 4. b)** Posterior probability density of the mean nitrous oxide flux from each land-use, estimated

by the Bayesian GLMM described in the text.

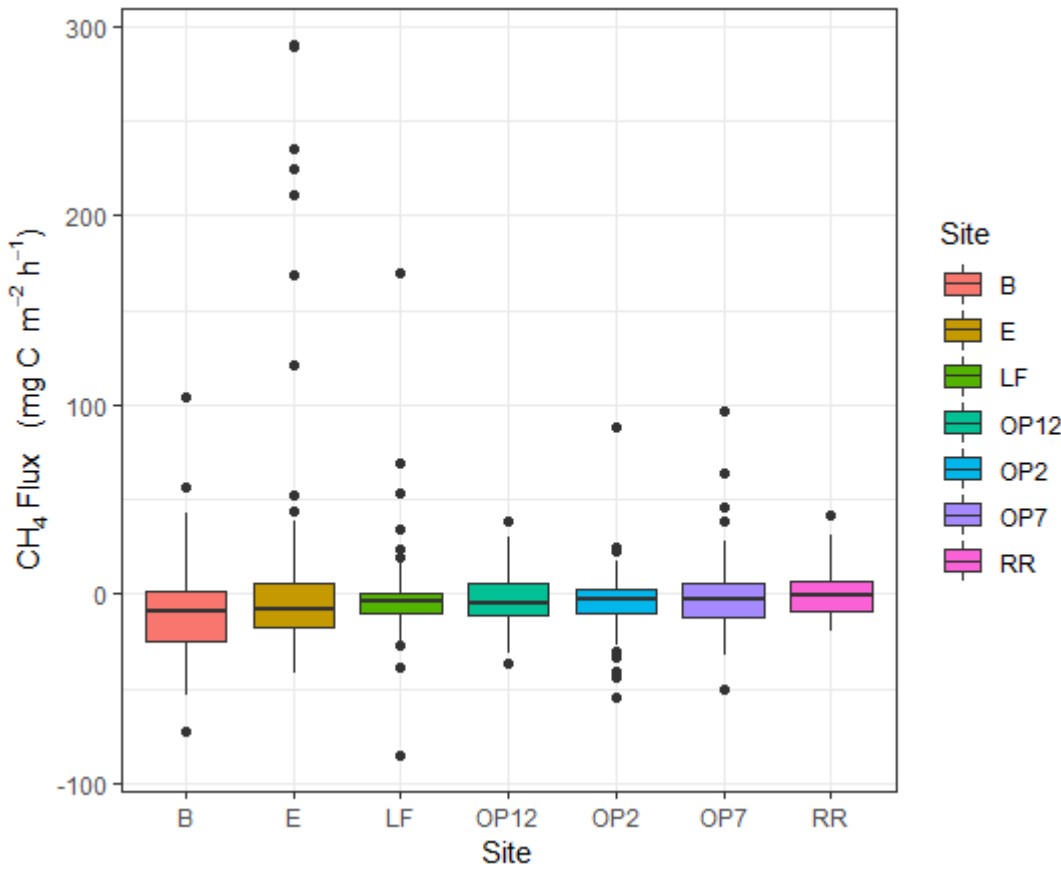

896

**Figure 5.** Methane ($CH_4$-C) fluxes in µg m$^{-2}$ h$^{-1}$ from the different sites from January 2015 - November 2016, every two months (B, E, LF = logged forests, OP12, OP2, OP7 = oil palm plantations, RR = riparian reserve). The ends of the box are the upper and lower quartiles, so the box spans the interquartile range. The median is marked by a horizontal line inside the box. The whiskers are the two lines outside the box that extend to the highest and lowest observations with outliers marked with an asterisk (*).



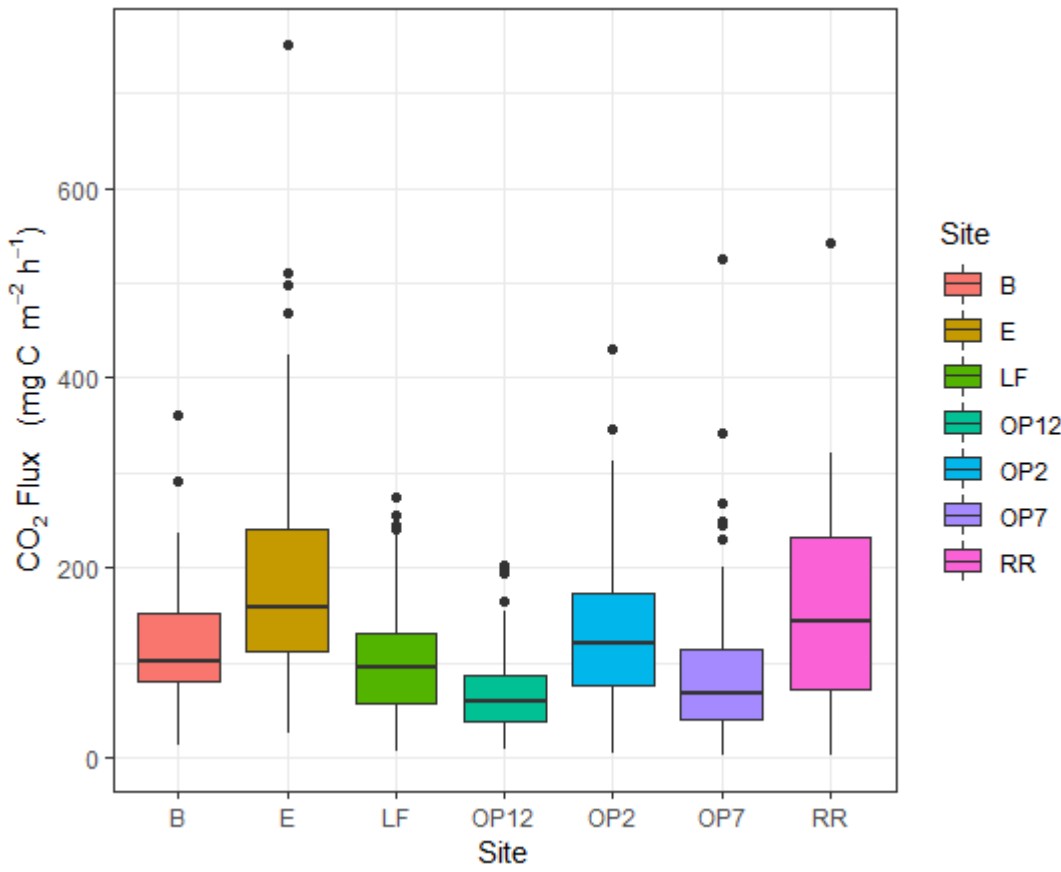

902

**Figure 6.** Soil respiration ($CO_2$-C) fluxes in mg $m^{-2}$ $h^{-1}$ from January 2015 - November 2016, every two months (B, E, LF = logged forests, OP12, OP2, OP7 = oil palm plantations, RR = riparian reserve). The ends of the box are the upper and lower quartiles, so the box spans the interquartile range. The median is marked by a horizontal line inside the box. The whiskers are the two lines outside the box that extend to the highest and lowest observations with outliers marked with an asterisk (*).







908

**Figure 7.** 2D Non metric multidimensional scaling ordination plots of bacteria, fungal and eukaryotic

communities from two samples dates March 2016 (upper panel, t1) and November 2016 (lower panel,

t2). Coloured points designate replicates from each site (B, E, LF = logged forests, OP12, OP2, OP7 =

oil palm plantations, RIP = riparian reserve), as indicated in the legend with additional site centroids

denoted on the plots. In addition, hulls indicate broad land-use categories as indicated in the legend.

914

915