# Peer review of "Comparison of greenhouse gas fluxes from tropical forests and oil"

_Biogeosciences, 2020_

## Referee Comment (RC1) · Anonymous Referee #1 · 29 Sep 2020

This manuscript by Drewer et al. examines how soil $N_2O$, $CO_2$, and $CH_4$ emissions differ between tropical forests and oil palm plantations. The authors show that mineral soil $N_2O$ emissions are higher from Oil Palm plantations compared to Forest or Riparian areas. Furthermore, the authors demonstrate that microbial communities differ between these land use types. The methods and data presented in this manuscript appear to be appropriate and support the author's main conclusions. To improve this manuscript, I suggest the authors better integrate this study in the context of their previously published study on the same system (Drewer et al. 2020) and improve their justification for measuring microbial community composition. My specific comments are detailed below.

General comments:

My main comment is that the authors do not adequately justify how total bacterial and fungal diversity will explain soil trace emissions. I suggest the authors use their previously published manuscript (Drewer et al. 2020) as justification for measuring microbial community composition in the introduction. Additionally, the authors only present total bacterial and fungal community composition data, and do not discuss which taxa contributed to the difference in microbial communities based on land use. By examining the known function of taxa that differ between land uses, the authors may be able to better link differences in microbial community composition to soil trace gas emissions. These data should be available based on the sequencing analyses described in the methods section.

Specific Comments:

Line 104-124: The authors do a nice job explaining how land use change might alter soil microbial communities in this paragraph. However, they do not justify how changes in bacterial/fungal diversity and richness might directly affect soil N2O emissions. To do this, the authors might consider presenting some of the main findings from their prior study (Drewer et al. 2020) in this paragraph.

Line 125 – 128: It would be useful if the authors presented hypotheses to accompany these objectives. This would help explain why they expect N2O emissions to vary based on land use and what soil properties they expect to drive these differences.

Line 155-156: I found the description of the study sites a little confusing in this paragraph. Here, what do "LF, B, and E" stand for?

Line 168 – 169: The authors should justify why they did not install equal number of chambers in each site.

Line 171-173: The authors should explain how they decided when to sample. Did they account for antecedent conditions such as time since last rain event?

Line 213 – 219: The authors did a nice job of describing how they calculated fluxes. However, if CO2 or N2O concentrations became saturated within the chamber headspace then linear regression could underestimate emissions – see Matthias et al. (1978) "A numerical evaluation of chamber methods for determining gas fluxes". The authors should discuss how they addressed this here.

Line 420: "CH4 oxidation" should be changed to "net CH4 oxidation"

Line 454-456: The authors should include these data in their ordination figure or as a table.

Line 603-608: I think this info could be incorporated into the previous paragraph.

Figure2: Here, and for all the figures, it would be useful if the OP sites were ordered from youngest to oldest.

Figure 3: The y-axis scale makes it difficult to see any patterns in the NH4 concentrations in the forest and riparian sites. I understand that this is to keep the axis scale consistent, but the authors should consider using a log scale or a broken y-axis so that the figures are easier to interpret. This is also true for figure 4.

Figure 5: It would be useful to see these data over time so we can see the presence or lack of any temporal patterns. This is also true for Figure 6.

---

## Referee Comment (RC2) · Anonymous Referee #2 · 29 Oct 2020

Drewer and colleagues present a manuscript about greenhouse gas emissions from tropical forest and oil palm plantation soils in the SABAH landscape of Southeast Asia. They compare emissions of N2O, CH4, and CO2 between the two different land use systems, want to upscale their results and try to find links between microbial communities and greenhouse gas fluxes.

I think you should exclude the whole microbial part of the manuscript to strengthen the greenhouse gas flux part. It is hard to see a substantial link between your results from microbial analysis and the greenhouse gas fluxes. The microbial part of the discussion remains very speculative because you are comparing diversity/composition with

greenhouse gas fluxes. It would have been better to have process rate measurements in the field (e.g. nitrification etc.) linked to functional gene abundance in soil samples where you had installed your static chambers (e.g. all the N-cycling genes, mcrA and/or pmoA). That would have been a sound story. Now you are reading two stories in one manuscript that do not strengthen each other.

The study design is the major drawback of the present manuscript. I do not understand why static chambers were not randomly installed. There was no plot selection as far as I can see. Why? There are only sites and per site you installed a different number of chambers (this is n=1) without any design!? How do you want to compare fluxes between land use systems if there are not enough replicates but only pseudoreplicates? How do you test differences of soil properties between the different land use systems? In my opinion the argument that Bayesian methodology is used to overcome the disadvantages of small sample size and high variability is very weak in your case. You could have easily selected few random plots within each site and then installed the same amount of chambers within each plot to overcome the different problems.

---

## Author Comment (AC1) · 12 Nov 2020

This manuscript by Drewer et al. examines how soil N2O, CO2, and CH4 emissions differ between tropical forests and oil palm plantations. The authors show that mineral soil N2O emissions are higher from Oil Palm plantations compared to Forest or Riparian areas. Furthermore, the authors demonstrate that microbial communities differ between these land use types. The methods and data presented in this manuscript appear to be appropriate and support the author's main conclusions. To improve this manuscript, I suggest the authors better integrate this study in the context of their pre-

viously published study on the same system (Drewer et al. 2020) and improve their justification for measuring microbial community composition. My specific comments are detailed below.

Response: We thank the reviewer for taking the time to read and comment on our manuscript. We reply to individual comments in turn below.

General comments: My main comment is that the authors do not adequately justify how total bacterial and fungal diversity will explain soil trace emissions.

Response: We will add additional references to the introduction to clarify this point. We measured both N2O and CH4 - fluxes which are the result of the activities of many different microbes – consumers and producers, therefore we just wish to examine if broad microbial metrics can help further explain flux variance.

I suggest the authors use their previously published manuscript (Drewer et al. 2020) as justification for measuring microbial community composition in the introduction. Additionally, the authors only present total bacterial and fungal community composition data, and do not discuss which taxa contributed to the difference in microbial communities based on land use. By examining the known function of taxa that differ between land uses, the authors may be able to better link differences in microbial community composition to soil trace gas emissions. These data should be available based on the sequencing analyses described in the methods section.

Response: We used the opportunity to investigate if there are changes in microbial community structure related to land-use change. We never tried to go beyond this (linking processes with specific taxa) as was done in the lab study. A detailed examination of specific microbial responses is beyond the scope of the current manuscript; and we prefer to present conclusions based on broad community metrics in the current MS.

Specific Comments: Line 104-124: The authors do a nice job explaining how land use

change might alter soil microbial communities in this paragraph. However, they do not justify how changes in bacterial/fungal diversity and richness might directly affect soil N2O emissions. To do this, the authors might consider presenting some of the main findings from their prior study (Drewer et al. 2020) in this paragraph.

Response: Fluxes are driven by many different microbes – producers and consumers and activities may be the result of multiple organismal interactions – this is why we focus on broad community metrics.

Line 125 – 128: It would be useful if the authors presented hypotheses to accompany these objectives. This would help explain why they expect N2O emissions to vary based on land use and what soil properties they expect to drive these differences.

Response: We will add following hypothesis to the MS: N2O fluxes will be larger from OP due to N fertiliser addition compared to tropical forest

Line 155-156: I found the description of the study sites a little confusing in this paragraph. Here, what do "LF, B, and E" stand for?

Response: We followed the design of the SAFE project and used their site names for consistency with other SAFE publications. We will add the land-use type as an extra column to table 1 and clarify in all tables and figures.

Line 168 – 169: The authors should justify why they did not install equal number of chambers in each site.

Response: We did install approximately equal numbers of chambers at each site, though not exact. To clarify the SAFE design, there were replicate sites for each land-use, with chambers installed at random locations within these sites. We had 3 forest sites each with 8 chambers randomly installed, 2 OP also with 8 randomly installed. The only deviation was OP7, where we tried to capture topography so had 12 chambers. The riparian was the outlier with only 4 but we don't focus on that in the land-use comparison. Our analysis does not assume a balanced design, and can cope with the

slight difference in the number of samples between sites.

Line 171-173: The authors should explain how they decided when to sample. Did they account for antecedent conditions such as time since last rain event?

Response: We tried to capture a 2 year period and had to take into account accessibility of the site so every two months was a pragmatic time scale that was achievable within the budget and allowing sample processing in between.

Line 213 – 219: The authors did a nice job of describing how they calculated fluxes. However, if $CO_2$ or $N_2O$ concentrations became saturated within the chamber headspace then linear regression could underestimate emissions – see Matthias et al. (1978) "A numerical evaluation of chamber methods for determining gas fluxes". The authors should discuss how they addressed this here.

Response: Fluxes have been quality checked and checked for linearity and no saturation occurred during the time sampled (2 min for $CO_2$ and 45 min for $N_2O$), so linear was the best fit for all fluxes presented here. We will add an extra sentence to clarify this.

Line 420: "$CH_4$ oxidation" should be changed to "net $CH_4$ oxidation"

Response: We will change the text accordingly.

Line 454-456: The authors should include these data in their ordination figure or as a table.

Response: We will add this to the supplementary information as it won't add much more to what is already in the text.

Line 603-608: I think this info could be incorporated into the previous paragraph.

Response: We will amend the text accordingly.

Figure2: Here, and for all the figures, it would be useful if the OP sites were ordered

from youngest to oldest.

Response: We will change the plots accordingly.

Figure 3: The y-axis scale makes it difficult to see any patterns in the NH4 concentrations in the forest and riparian sites. I understand that this is to keep the axis scale consistent, but the authors should consider using a log scale or a broken y-axis so that the figures are easier to interpret. This is also true for figure 4.

Response: As there are no real patterns, we decided to use this format but can replot the figures using a broken axis.

Figure 5: It would be useful to see these data over time so we can see the presence or lack of any temporal patterns. This is also true for Figure 6.

Response: As there are no temporal patterns, we decided to use this format but for consistency can replot these figures the same way as figure 4a.

---

## Author Comment (AC2) · 12 Nov 2020

Anonymous Referee #2 Drewer and colleagues present a manuscript about greenhouse gas emissions from tropical forest and oil palm plantation soils in the SABAH landscape of Southeast Asia. They compare emissions of N2O, CH4, and CO2 between the two different land use systems, want to upscale their results and try to find links between microbial communities and greenhouse gas fluxes.

Response: We thank the reviewer for taking the time to read and comment on our manuscript. We reply to individual comments in turn below.

I think you should exclude the whole microbial part of the manuscript to strengthen the greenhouse gas flux part. It is hard to see a substantial link between your results from microbial analysis and the greenhouse gas fluxes. The microbial part of the discussion remains very speculative because you are comparing diversity/composition with greenhouse gas fluxes. It would have been better to have process rate measurements in the field (e.g. nitrification etc.) linked to functional gene abundance in soil samples where you had installed your static chambers (e.g. all the N-cycling genes, mcrA and/or pmoA). That would have been a sound story. Now you are reading two stories in one manuscript that do not strengthen each other.

Response: We could have done many other things but our goal here was to both describe how the treatments affect broad microbial communities, but also use this data in the predictive models. We still believe the microbial data helps in identifying differences between sites and use the microbial metrics in the models to try and explain fluxes.

The study design is the major drawback of the present manuscript. I do not understand why static chambers were not randomly installed. There was no plot selection as far as I can see. Why? There are only sites and per site you installed a different number of chambers (this is n=1) without any design!? How do you want to compare fluxes between land use systems if there are not enough replicates but only pseudoreplicates? How do you test differences of soil properties between the different land use systems?

Response: The referee misunderstands the sampling design we used, and we have clarified this in the manuscript. We did install approximately equal numbers of chambers at each site, though this was not exact. To clarify the SAFE design, there were replicate sites for each land-use, with chambers installed at random locations within these sites. We had 3 forest sites each with 8 chambers, and 3 OP sites with 8 or more chambers. The only anomaly was OP7, where we tried to capture topographic variation, so had 12 chambers. The riparian was the outlier with only 4 but we don't focus on that in the land-use comparison. Our statistical analysis does not assume a balanced design, and can cope with the slight difference in the number of samples between sites.

[Figure]

We did not aim to test for differences in soil properties between the different land use systems, only whether these could explain the variability in N2O fluxes.

In my opinion the argument that Bayesian methodology is used to overcome the disadvantages of small sample size and high variability is very weak in your case.

Response: The methodology does not overcome the problem of small sample size and high variability, but makes the uncertainty associated with this very clear, by characterising the posterior probability distribution properly. It is thereby an appropriate method to use in this context.

You could have easily selected few random plots within each site and then installed the same amount of chambers within each plot to overcome the different problems.

Response: See response above, this is basically what has been done with one exception.

---

## Author Response (AR1)

Associate Editor Decision: Reconsider after major revisions (14 Nov 2020) by Andreas Ibrom
Comments to the Author:

Dear Authors,

thank you for your final responses to the reviews.

There are two major points of critique, one being the sampling scheme and the other the missing link between the observations of microbial community composition and the biogeochemical processes and GHG fluxes. While it looks as if you are able to address the critique on the sampling scheme and the Bayesian analysis by a more accurate and clear description, you seem to dismiss the idea of the two reviewers of better linking the processes to microbial community composition. I would like to make you aware of that both reviewers have expressed their dissatisfaction on the issue and it is finally their evaluation that I am bound to follow.

With this said, I am confident that you'll find a more positive way to respond to this issue raised by the two referees and, by that, making the work more concise, novel and increasing its scientific impact.

With kind regards,
Andreas Ibrom

Reply:
Dear Andreas,

Thank you for your helpful suggestions. We have edited the text throughout the manuscript to address the concerns raised and respond to individual points in turn below.

We have clarified that the Bayesian analysis can cope with a slight imbalance in the sampling design and small sample size by appropriately characterising the uncertainties.

Regarding the microbial community indices used, we now clarify throughout the MS that we only used this as an additional variable in the GLMM (in addition to other abiotic parameters) in order to try and explain the variance in measured $N_2O$ fluxes. We did not aim to identify processes driving $N_2O$ fluxes. Throughout the manuscript we have toned down the part of the microbial analyses and changed the title to "Comparison of greenhouse gas fluxes from tropical forests and oil palm plantations on mineral soil" removing the mention of the microbial work, as this is such a small part in our paper.

To further clarify we have now added following hypotheses:
    (1) $N_2O$ fluxes will be larger from OP plantations due to N fertiliser addition compared to tropical forests.
    (2) Land use determines microbial diversity, and thereby influence $N_2O$ flux rates.

We hope that the changes we made sufficiently address the concerns raised by the editor and the reviewers.

With kind regards,
Julia Drewer

Anonymous Referee #1

This manuscript by Drewer et al. examines how soil N2O, CO2, and CH4 emissions differ between tropical forests and oil palm plantations. The authors show that mineral soil N2O emissions are higher from Oil Palm plantations compared to Forest or Riparian areas. Furthermore, the authors demonstrate that microbial communities differ between these land use types. The methods and data presented in this manuscript appear to be appropriate and support the author's main conclusions. To improve this manuscript, I suggest the authors better integrate this study in the context of their previously published study on the same system (Drewer et al. 2020) and improve their justification for measuring microbial community composition. My specific comments are detailed below.

Response: We thank the reviewer for taking the time to read and comment on our manuscript. Throughout the manuscript we have now added clarification on the intention of using the microbial community data as just another variable in addition to the abiotic parameters to be used in the GLMM. We reply to individual comments in turn below.

General comments: My main comment is that the authors do not adequately justify how total bacterial and fungal diversity will explain soil trace emissions.

Response: Regarding the microbial community indices used, we now clarify throughout the MS that we only used this as an additional variable in the GLMM (in addition to other abiotic parameters) in order to try and explain the variance in measured $N_2O$ fluxes. We did not aim to identify processes driving $N_2O$ fluxes. Throughout the manuscript we have toned down the part of the microbial analyses and changed the title to "Comparison of greenhouse gas fluxes from tropical forests and oil palm plantations on mineral soil" removing the mention of the microbial work.

I suggest the authors use their previously published manuscript (Drewer et al. 2020) as justification for measuring microbial community composition in the introduction. Additionally, the authors only present total bacterial and fungal community composition data, and do not discuss which taxa contributed to the difference in microbial communities based on land use. By examining the known function of taxa that differ between land uses, the authors may be able to better link differences in microbial community composition to soil trace gas emissions. These data should be available based on the sequencing analyses described in the methods section.

Response: To expand on the point above, we simply seek to use broad microbial community metrics as additional explanatory variables for fluxes and have toned down the focus throughout the MS. We have reduced emphasis on microbial community aspects in light of reviewers concerns and changed the title accordingly. We used the opportunity to investigate if there are changes in microbial community structure related to land-use and never intended to go beyond.
To clarify our position we have added the following paragraph at the end of the introduction section (Line 113-120):

'Although the focus of this paper lies on the comparison of soil GHG flux rates (especially for $N_2O$) and their soil chemical and physical properties, we have taken the opportunity to understand the differences in microbial community composition between forests and OP in situ. A previous study has investigated environmental drivers and microbial pathways leading to GHG emissions under controlled laboratory incubations using soils from a subset of the field locations discussed here (Drewer et al., 2020). The aim here was to broadly characterise the microbial communities at the different sites in the different land-uses and use the information alongside other measured abiotic factors in mixed models in an attempt to explain the measured fluxes.'

And added following hypothesis:

(2)	Land-use determines microbial diversity, and thereby influences N$_2$O flux rates

In addition, we revised relevant sections in the discussion section (Line 601-606)

'To what extend these differences impact on microbial processes leading to GHG fluxes is hardly known (Kaupper et al., 2019). Despite our data showing effects of land-use and soil properties on components of the microbial communities (fungal and eukaryote), including of microbial community metrics in the GLMM did not help to explain variability in N$_2$O fluxes. Hence, we partially prove our hypothesis that microbial diversity is determined by land-use but have to disprove the latter part of the second hypothesis (microbial diversity did not influence N$_2$O fluxes).'

Specific Comments:

Line 104-124: The authors do a nice job explaining how land use change might alter soil microbial communities in this paragraph. However, they do not justify how changes in bacterial/fungal diversity and richness might directly affect soil N2O emissions. To do this, the authors might consider presenting some of the main findings from their prior study (Drewer et al. 2020) in this paragraph.

Response:  Please see response to previous points. We have clarified the focus and intention of using microbial communities, i.e. only as additional variable in the GLMM, throughout.

Line 125 – 128: It would be useful if the authors presented hypotheses to accompany these objectives. This would help explain why they expect N2O emissions to vary based on land use and what soil properties they expect to drive these differences.

Response: We have added following hypotheses to lines 129-131:

(1)	N$_2$O fluxes will be larger from OP plantations due to N fertiliser addition compared to tropical forest

(2)	Land-use determines microbial diversity, and thereby influences N$_2$O flux rates

Line 155-156: I found the description of the study sites a little confusing in this paragraph. Here, what do "LF, B, and E" stand for?

Response: We followed the design of the SAFE project and used their site names for consistency with other SAFE publications, and added following sentences in lines 150-159:

'To be consistent with previous and future SAFE publications, we use the site labelling as per the SAFE convention, detailed below. … We selected a young OP plantation, around 2 years old at the time we started measurements (OP2) and a medium aged OP plantation, around 7 years old at the start of the project (OP7). The riparian reserve area (RR), draining into a small shallow stream, is adjacent and down slope from OP7. In addition, we selected a slightly older plantation, around 12 years of age at the start of the project (OP12). All OP plantations in this study were terraced. Logged forest sites are the 10 ha plots of the logged forest (and future fragments) LF, B and E of the SAFE design.'

We also clarified land-use type and site names in all table headings and all figure captions.

Line 168 – 169: The authors should justify why they did not install equal number of chambers in each site.

Response: We did install equal numbers of chambers at all but 2 sites (RR and OP7). To clarify the SAFE design, there were replicate sites for each land-use, with chambers installed at random locations within these sites. We had 3 forest sites each with 8 chambers randomly installed, 2 OP also with 8 randomly installed. The only deviation was OP7, where we tried to capture variability regarding topography, so had 12 chambers. However, there was no variability in terms of fluxes and soil properties, so the 12 sample locations were analysed as together. The riparian area was not easy to access, for this reason only 4 chambers were installed. Our statistical analysis does not assume a

balanced design, and can cope with the slight difference in the number of samples between sites. This has been clarified in section 2.2.

Line 171-173: The authors should explain how they decided when to sample. Did they account for antecedent conditions such as time since last rain event?
Response: We tried to capture a 2 year period and had to take into account accessibility of the site so every two months was a pragmatic time scale that was achievable within the budget and allowing sample processing in between. The SAFE site is remote (a day's travel by plane and jeep). Rainfall events are almost daily and very localised. There is only one rain collector at the SAFE field camp, and this collector is not necessarily representative of the individual sites, which are spread across a large area of 8,000 ha (https://www.safeproject.net/info/design). It is therefore not possible to account for antecedent conditions.
In the method section (Line 170-171) we included the scale of the SAFE region: 'In order to measure fluxes of $N_2O$ and $CH_4$ from the chosen logged forests and OP plantations, a total of 56 static chambers were installed in the SAFE landscape (total area 8,000 ha).'

Line 213 – 219: The authors did a nice job of describing how they calculated fluxes. However, if CO2 or N2O concentrations became saturated within the chamber headspace then linear regression could underestimate emissions – see Matthias et al. (1978) "A numerical evaluation of chamber methods for determining gas fluxes". The authors should discuss how they addressed this here.
Response: Fluxes have been quality checked and checked for linearity and no saturation occurred during the time sampled (2 min for $CO_2$ and 45 min for $N_2O$ and $CH_4$), so linear was the best fit for all fluxes presented here. We have added the following sentence in line 229-230 to clarify this.
'Fluxes were quality checked and checked for linearity and no saturation occurred during the time sampled (2 min for $CO_2$ and 45 min for $N_2O$ and $CH_4$), so linear was the best fit for all fluxes presented here.'

Line 420: "CH4 oxidation" should be changed to "net CH4 oxidation"
Response: We have changed the text accordingly.

Line 454-456: The authors should include these data in their ordination figure or as a table.
Response: We added Supplementary Table 1 with the data as requested.

Line 603-608: I think this info could be incorporated into the previous paragraph.
Response: We have removed the entire paragraph in a substantial re-write of the discussion section.

Figure2: Here, and for all the figures, it would be useful if the OP sites were ordered from youngest to oldest.
Response: We have changed all figures accordingly.

Figure 3: The y-axis scale makes it difficult to see any patterns in the NH4 concentrations in the forest and riparian sites. I understand that this is to keep the axis scale consistent, but the authors should consider using a log scale or a broken y-axis so that the figures are easier to interpret. This is also true for figure 4.
Response: We have replotted all figures using different axes scales to visualise the differences better and have noted this in figure captions as appropriate.

Figure 5: It would be useful to see these data over time so we can see the presence or lack of any temporal patterns. This is also true for Figure 6.
Response: We have changed figures 5 and 6 accordingly and have moved the old figures to the Supplementary Information as they illustrate the range of measured fluxes in addition to the temporal variability.

Anonymous Referee #2
Drewer and colleagues present a manuscript about greenhouse gas emissions from tropical forest and oil palm plantation soils in the SABAH landscape of Southeast Asia. They compare emissions of N2O, CH4, and CO2 between the two different land use systems, want to upscale their results and try to find links between microbial communities and greenhouse gas fluxes.
Response: We thank the reviewer for taking the time to read and comment on our manuscript. We reply to individual comments in turn below.

I think you should exclude the whole microbial part of the manuscript to strengthen the greenhouse gas flux part. It is hard to see a substantial link between your results from microbial analysis and the greenhouse gas fluxes. The microbial part of the discussion remains very speculative because you are comparing diversity/composition with greenhouse gas fluxes. It would have been better to have process rate measurements in the field (e.g. nitrification etc.) linked to functional gene abundance in soil samples where you had installed your static chambers (e.g. all the N-cycling genes, mcrA and/or pmoA). That would have been a sound story. Now you are reading two stories in one manuscript that do not strengthen each other.
Response: In line with comments also from reviewer #1 and the editor we have substantially rewritten the manuscript toning down the part of the microbial communities and changed the title removing the mention of microbial communities. The intention was to use the microbial community data as just another variable in addition to the abiotic parameters to be used in the GLMM.
We have reduced emphasis on microbial community aspects in light of reviewers concerns and changed the title accordingly. We used the opportunity to investigate if there are changes in microbial community structure related to land-use and never intended to go beyond.
To clarify our position we have added the following paragraph at the end of the introduction section (Line 113-120):

'Although the focus of this paper lies on the comparison of soil GHG flux rates (especially for N2O) and their soil chemical and physical properties, we have taken the opportunity to understand the differences in microbial community composition between forests and OP in situ. A previous study has investigated environmental drivers and microbial pathways leading to GHG emissions under controlled laboratory incubations using soils from a subset of the field locations discussed here (Drewer et al., 2020). The aim here was to broadly characterise the microbial communities at the different sites in the different land-uses and use the information alongside other measured abiotic factors in mixed models in an attempt to explain the measured fluxes.'

And added following hypothesis:
(2)    Land-use determines microbial diversity, and thereby influences N2O flux rates

In addition we revised relevant sections in the discussion section (Line 601-606)
'To what extend these differences impact on microbial processes leading to GHG fluxes is hardly known (Kaupper et al., 2019). Despite our data showing effects of land-use and soil properties on components of the microbial communities (fungal and eukaryote), including of microbial community metrics in the GLMM did not help to explain variability in N2O fluxes. Hence, we partially prove our

hypothesis that microbial diversity is determined by land-use but have to disprove the latter part of the second hypothesis.'

The study design is the major drawback of the present manuscript. I do not understand why static chambers were not randomly installed. There was no plot selection as far as I can see. Why? There are only sites and per site you installed a different number of chambers (this is n=1) without any design!? How do you want to compare fluxes between land use systems if there are not enough replicates but only pseudoreplicates? How do you test differences of soil properties between the different land use systems?

Response: The reviewer perhaps misunderstands the sampling design we used, and we have clarified this in the manuscript, see also the address to reviewer 1 on the same issue and shown below.

We did install equal numbers of chambers at all but 2 sites (RR and OP7). To clarify the SAFE design, there were replicate sites for each land-use, with chambers installed at random locations within these sites. We had 3 forest sites each with 8 chambers randomly installed, 2 OP also with 8 randomly installed. The only deviation was OP7, where we tried to capture variability regarding topography, so had 12 chambers. However, there was no variability in terms of fluxes and soil properties, so the 12 sample locations were analysed as together. The riparian area was not easy to access, for this reason only 4 chambers were installed. Our statistical analysis does not assume a balanced design, and can cope with the slight difference in the number of samples between sites.

We did not aim to test for differences in soil properties between the different land use systems, only when these could explain the variability in $N_2O$ fluxes.

We have modified the methods section l 330-339 for clarification: 'Here we applied a Bayesian methodology to address this problem, using a model similar to that described by Levy et al. (2017). This accounts for the lognormal distribution of observations, while including hierarchical effects of land-use, and effects of sites within land-use types as well as the repeated measures. In the current statistical terminology, this is a generalised linear mixed-effect model (GLMM) with a lognormal response and identity link function. The model consists of a fixed effect of land-use (Forest, Oil Palm, or Riparian), with a random effect representing the variation among sites within a land-use type. The parameters were estimated by the Markov chain Monte Carlo (MCMC) method, using Gibbs sampling as implemented in Just Another Gibbs Sampler (JAGS) (Plummer 1994), and described in more detail by Levy et al. (2017). The model can cope with the slight imbalance in the design, and propagates the uncertainty associated with the relatively small sample sizes appropriately.'

In my opinion the argument that Bayesian methodology is used to overcome the disadvantages of small sample size and high variability is very weak in your case.

Response: The methodology does not overcome the problem of small sample size and high variability, but makes the uncertainty associated with this very clear, by characterising the posterior probability distribution properly. It is thereby an appropriate method to use in this context.

We have clarified the text in Line 330-339, see comment above.

You could have easily selected few random plots within each site and then installed the same amount of chambers within each plot to overcome the different problems.

Response: This is basically what has been done with exception of OP 7 (12 chambers) and RR (4 chambers). See response above addressing your previous and reviewer 1 comments.